# Effect of Different Lead and Cadmium Salts on the Photolytic Degradation of Two Typical Fluoroquinolones under Natural Sunlight Irradiation

**DOI:** 10.3390/ijerph20010323

**Published:** 2022-12-25

**Authors:** Lunchao Duan, Hao Yang, Fenghe Wang

**Affiliations:** 1School of Environment, Nanjing Normal University, Nanjing 210023, China; 2Jiangsu Province Science and Technology Resources Coordination and Service Center, Nanjing 210018, China; 3School of Geography, Nanjing Normal University, Nanjing 210023, China; 4Key Laboratory for Soft Chemistry and Functional Materials of Ministry of Education, Nanjing University of Science and Technology, Nanjing 210097, China

**Keywords:** photolytic degradation, levofloxacin, norfloxacin, Pb^2+^/Cd^2+^, NO_3_^−^/Cl^−^

## Abstract

This study investigated the effects of different lead and cadmium salts (Pb(NO_3_)_2_, Cd(NO_3_)_2_, PbCl_2_, and CdCl_2_) on the photolytic degradation of two typical fluoroquinolones (levofloxacin (LVF) and norfloxacin (NOR)) under natural sunlight irradiation. Their half-life time and photolytic kinetic constants (*k*) were calculated at different molar ratios. The results indicated that the photolytic degradation curves of LVF and NOR followed apparent first-order kinetics. After 42 days of sunlight irradiation, approximately 48.3–69.4% of NOR was decomposed when the initial concentration increased from 0.006 to 0.06 mmol/L. In comparison, only 9.8–43.4% of LVF was decomposed. The *k* of NOR ranged from 0.79 × 10^−3^ to 1.30 × 10^−3^ h^−1^, and the *k* of LVF increased from 6.82 × 10^−4^ to 1.61 × 10^−4^ h^−1^. Compared with the control, the Pb^2+^ and Cd^2+^ participation tended to enhance the LVF and NOR photodegradation. The effects of Cd^2+^ on the photodegradation efficiency were more significant than those of Pb^2+^. It was inferred that the presence of aqueous NO_3_^−^ obviously suppressed the NOR degradation, but Cl^−^ had slight effects on these two fluoroquinolones’ photodegradation. These results are of importance toward the understanding of the persistence of FQs under natural sunlight irradiation in surface waters.

## 1. Introduction

Fluoroquinolones (FQs), such as norfloxacin (NOR) and levofloxacin (LVF), are a large class of antibiotics that are widely used in aquaculture, livestock husbandry, and human prescription for their broad activity spectrum and good oral intake properties [1,2]. Therefore, the widespread detection of FQs in terrestrial and aquatic systems has engendered significant scientific and regulatory concern. Previous studies showed that FQs are some of the most frequently detected antibiotics in surface ecosystems [3,4]. Relatively high residual FQ concentrations (ng/L to several μg/L) have been widely detected in seas [5,6], rivers [7,8], lakes [9,10], groundwater [11,12], and wastewater [13,14]. Ciprofloxacin (0–3.40 μg /L), norfloxacin (0.037–9.35 μg /L), and ofloxacin (0.047–8.64 μg /L) are the most commonly detected quinolone antibiotics in effluents [15], which could potentially pose a significant risk to ecosystems and human health.

The attenuation of FQs in the aquatic environment is mainly attributed to their biodegradation, photolytic degradation, and chemical degradation [2,16]. Of these degradation methods, photolytic degradation plays a significant role in FQs’ fate in some natural waters [17,18,19,20,21]. FQs can directly absorb sunlight and undergo photolysis, including direct photolysis and self-sensitized photooxidation via reactive oxygen species (ROS) such as hydroxyl radicals (•OH) and singlet oxygen (^1^O_2_). Moreover, the water constituents may also affect the photolytic degradation of FQs. Some studies have focused on the existing anions in the photodegradation of FQs [17,22]. In addition, the impact of coexisting metal cations on the environmental behaviors of FQs is a great of interest [17,23,24,25,26]. The co-existence of FQs with metal cations can form stable complexes and therefore this complexation interaction may alter FQs’ properties. Sciscenko et al. [26] found that iron complexation notably diminished enrofloxacin’s photodegradation. Ge et al. [17] also proved that Fe^3+^ inhibited the photodegradation of FQs. A possible reason was that Fe ions act as radiation filters and/or scavengers for ROS, which therefore inhibit the photolysis and/or photooxidation of FQs. With the development of modern agriculture and industry, increasing amounts of heavy metals along with FQs have been continuously discharged into surface water environments [9,27,28,29]. As reported in a survey in China, the total emissions of heavy metal(loid)s such as lead, cadmium, chromium, and arsenic in national wastewater reached 120 tons in 2019. Many studies have shown that rivers, surface water, and sewage contain heavy metals, in the range of ng/L~μg/L, such as Cd (0.025 μg/L) in Taihu Lake [30], Pb (1.21 μg/L) in the Yangtze River, and Pb (0.68 μg/L) Cd (0.07 μg/L) in the urban surface water of Su Zhou. Vinod Kumar et al. [31] collected 147 publications and proceedings on heavy metals in worldwide surface water bodies and found that the average concentration of Cd was 180 μg/L. It is necessary to investigate the environmental fate of FQs when they interact with heavy metals, particularly when the complex systems are exposed to natural sunlight in actual environmental scenarios. Nevertheless, research on this topic still remains limited and it needs further investigation.

As a result, this study focused on characterizing the influence of natural sunlight on the photodegradation of two typical FQs (LVF and NOR) with the interference of different Pb and Cd salts (Pb(NO_3_)_2_, PbCl_2_, Cd(NO_3_)_2_, and CdCl_2_) at different molar ratios. The results could facilitate an understanding of the FQs’ photolytic degradation in aquatic environments.

## 2. Materials and Methods

### 2.1. Materials and Chemicals

NOR was supplied by Hangzhou Minsheng Pharmaceutical Co., Ltd. (Hangzhou, China). LVF was obtained from Guangdong Eashu Pharmaceutical Co., Ltd. (Zhaoqing, China). Pb(NO_3_)_2_, Cd(NO_3_)_2_, PbCl_2_, and CdCl_2_ were purchased from Aladdin Industrial Co., Ltd. (Hong Kong, China). All the reagents were used without further purification.

### 2.2. Standard Curve of FQs

The FQ concentrations in the aqueous samples were measured using high-pressure liquid chromatography (HPLC, P1201, Dalian Elite Analytical Instruments Co., Ltd., Dalian, China) with a UV–vis detector and a UV–vis spectrophotometer (UV-2550, Shimadzu, Kyoto, Japan). The absorbance values of NOR and LVF were recorded at 273 and 291 nm wavelengths, respectively (Appendix A). The standard concentrations of NOR and LVF (2.0–20 mg/L) were prepared by diluting the stock solution with deionized water. The calibration plot of NOR and LVF absorbance versus their concentrations shows a linear variation, with a correlation coefficient (*R*^2^) higher than 0.999 (Appendix A), indicating the reliable determination of this method.

### 2.3. Determination of Complexation Ratio

Complex formation between Pb/Cd ions and NOR/LVF was investigated using the molar ratio method by UV–vis spectroscopy. A 25 mL stock solution of FQs was transferred to a volumetric flask at 25 °C, and then a certain dosage of Pb^2+^/Cd^2+^ was added, causing the concentration ratio (C_NOR/LVF_/C_Pb/Cd_) to range from 0.5 to 5. The absorbance of the complex solutions was plotted versus the concentration ratio (C_NOR/LVF_/C_Pb/Cd_), and thus the concentration ratio curves were obtained.

### 2.4. Photolytic Degradation of FQs with or without Metal Salts

A series of NOR and LVF solutions (0.006, 0.012, and 0.06 mmol/L) were prepared by diluting the stock solution with deionized water. In the photodegradation experiments, 1 L of each solution was transferred to a 5 L beaker and all the beakers were placed in a glass box under natural sunlight irradiation. The experiment was conducted in June and July 2020, with an average temperature of 20 °C. At certain time intervals (ranging from 0 to 936 h), an aliquot of the solution was collected for the measurement. In addition, the Pb(NO_3_)_2_, Cd(NO_3_)_2_, PbCl_2_, and CdCl_2_ were added to different concentrations of NOR and LVF solutions at molar ratios (Pb/Cd to NOR/LVF) of 1:1, 1:2, and 1:3, respectively. Control tests were also conducted without Pb or Cd salt addition. The photodegradation experiments were the same as those described above. Each treatment and the control were prepared with three replicates.

The NOR or LVF concentrations in the sampling aliquots were analyzed through a UV–vis spectrophotometer and HPLC. All the analysis were conducted in triplicate, and the results reported in this study are the average values. After each measurement, the collected aliquot was returned to the corresponding beaker. Deionized water was supplemented at regular intervals to maintain the beakers at constant weights at the beginning of experiments.

## 3. Results and Discussion

### 3.1. Complexation Ratio Determination

The complexation ratio of NOR/LVF with Pb^2+^ and Cd^2+^ was determined by using UV–vis spectroscopy. As shown in Appendix A, the complexation ratio between FQs and Pb^2+^ or Cd^2+^ was around 1:2, which was consistent with previous studies [25]. Different heavy metal cations and molar ratios with FQs mean that different complexation structures will be formed, and their complexation structure may affect its photolytic degradation [25].

### 3.2. Effects of Initial FQ Concentrations

Figure 1 shows the effects of the initial concentrations (0.006, 0.012, and 0.06 mmol/L) on the photolytic degradation of NOR and LVF at different time intervals. The obtained pseudo-first-order kinetic parameters (correlation coefficients (*R*^2^), rate constants (*k*), half-life (*t*_1/2_)) and the degradation efficiencies (%) of FQs are presented in Appendix A. The results indicated that the photolytic degradation of NOR followed the pseudo-first-order kinetics, with *R*^2^ values higher than 0.95 for all concentrations (Appendix A). The photolytic degradation of LVF followed the pseudo-first-order kinetics at lower concentrations (0.006 and 0.012 mmol/L), with *R*^2^ values higher than 0.91 (Appendix A). In general, our results were consistent with many previous studies, reporting that the FQs’ photodegradation followed apparent first-order kinetics when exposed to sunlight irradiation [17,20,32]. With an increase in the initial FQ concentration, a decrease trend in the photodegradation rate constant can be observed, particularly for LVF. After 42 days of sunlight irradiation, approximately 57.2%, 69.4%, and 48.3% of NOR was decomposed for 0.006, 0.012, and 0.06 mmol/L, respectively. In comparison, only 43.4%, 30.3%, and 9.8% of LVF was decomposed for 0.006, 0.012, and 0.06 mmol/L, respectively. Moreover, the rate constants of NOR were higher than those of LVF, and, correspondingly, the half-life values of NOR were smaller than those of LVF. These results indicated that the photodegradation ability of NOR was relatively higher than that of LVF.

### 3.3. Effects of Lead and Cadmium Salts on the Photolytic Degradation of Two FQs

The effects of Pb and Cd salts on FQs’ photolytic degradation at different molar ratios are shown in Figure 2, Figure 3, Figure 4 and Figure 5. In all of the studied cases, the photolytic degradation data fitted the pseudo-first order model (lnc = lnc_0_ − kt), and the half-life times and decomposition rates for all samples are listed in Appendix A.

As shown in Figure 2 and Figure 3, the FQs were prone to photolytic degradation under natural sunlight irradiation (42 days) without the addition of Pb or Cd salts, and NOR was more easily attenuated (69.4%) compared to LVF (30.3%). When the nitrate was mixed, the NOR removal efficiency ranged from 57.3% to 61.8% with Pb^2+^ as the positive ion, and it ranged from 63.4% to 67.3% with Cd^2+^ as the positive ion (Figure 2). When the chloride was mixed, the NOR removal efficiency changed to 67.5–71.4% with Pb^2+^ as the positive ion, and to 65.5–72.7% with Cd^2+^ as the positive ion (Figure 2). When the nitrate was mixed, the LVF removal efficiency ranged from 33.0% to 37.5% with Pb^2+^ as the positive ion, and ranged from 38.2% to 46.1% with Cd^2+^ as the positive ion (Figure 3). When the chloride was mixed, the LVF removal efficiency increased to 37.4–47.8% with Pb^2+^ as the positive ion, and to 47.5–58.3% with Cd^2+^ as the positive ion (Figure 3). In general, the removal efficiencies of both NOR and LVF were higher in FQs-Cd^2+^ systems than those in FQ-Pb^2+^ systems at each molar ratio. As shown in Appendix A, for the same anion (NO_3_^−^ or Cl^−^), the *t*_1/2_ values in FQ-Cd^2+^ systems were generally smaller than those in FQ-Pb^2+^ systems, and this phenomenon was more evident for LVF.

The effects of NO_3_^−^ and Cl^−^ on the photolytic degradation efficiency of the FQs are shown in Figure 4 and Figure 5. As shown in Figure 4, when the cations (Pb^2+^ or Cd^2+^) remained unchanged, the final removal rates of NOR tended to be higher when the anion was Cl^−^, whereas they were lower when the anion was NO_3_^−^, compared to the control test. In addition, this phenomenon was more evident at lower molar ratios (1:2 and 1:3). Moreover, the degradation rates of NOR in NOR-Cl^−^ systems were faster in the first half of the reaction (until approximately 500 h). Similarly, when the cations (Pb^2+^ or Cd^2+^) remained unchanged, the final removal rates of LVF were higher when the anion was Cl^−^ in comparison with NO_3_^−^. Moreover, this phenomenon was most obvious when the molar ratio was 1:2 for Pb^2+^ and 1:3 for Cd^2+^ (Figure 5). These results indicated that Cl^−^ greatly accelerated the degradation of both NOR and LVF in the aqueous solution, without regard to the cations (Pb^2+^ or Cd^2+^). As shown in Appendix A, for the same cation (Pb^2+^ or Cd^2+^), the *t*_1/2_ values of the FQ-NO_3_^−^ systems were higher than those in the FQ-Cl^−^ systems at each molar ratio. Almost all the *t*_1/2_ values of NOR-NO_3_^−^ systems were higher than 22 days (the *t*_1/2_ of NOR blank sample), implying that NO_3_^−^ retarded NOR degradation under natural sunlight irradiation. In comparison, almost every *t*_1/2_ of the NOR-Cl^−^ system was less than 22 days, reflecting that Cl^−^ tended to enhance the degradation of NOR. Meanwhile, for the LVF, all the *t*_1/2_ values of the LVF-NO_3_^−^ and LVF-Cl^−^ systems were obviously less than 65 days (the *t*_1/2_ of the blank sample of LVF), indicating that both nitrate and chloride played a positive role in the degradation of LVF. Moreover, each *t*_1/2_ value of the LVF-NO_3_^−^ system was higher than that of the LVF-Cl^−^ system at the same molar ratio.

The co-existence of FQs with metal ions in the aqueous environment leads to metal complexation, and this interaction may alter FQs’ properties. Previous studies reported the inconsistent effects of metal ions on FQs’ photodegradation efficiency. The photodegradation rate of RFX (a frequently found FQ) decreased in the presence of metal ions, but no changes occurred in the nature of the photoproducts [33]. The Cu(II)-CIP (a typical FQ) complexation inhibited the photodegradation of CIP and altered the photolytic pathways and products [34]. However, it was reported that metal ions (such as Cu, Zn, Fe, and Al) enhanced the photodegradation of moxifloxacin (a typical FQ) in aqueous solutions [35]. These findings indicate that the effect of the complexation reaction on FQs’ photodegradation greatly depends on the antibiotic molecular structure and metal type. Our results indicated that both Pb^2+^ and Cd^2+^ tend to enhance the photodegradation of these two FQs, and the promoting effect of Cd^2+^ was more significant than that of Pb^2+^, particularly for LVF. However, the effect of anions on the comprehensive photodegradation efficiency could not be neglected.

As reported previously, the presence of NO_3_^−^ and Cl^−^ might affect the photodegradation of FQs in aqueous solutions, but the findings were not unitary. Some studies proved that NO_3_^−^ and Cl^−^ decreased the photodegradation rates of FQs under VUV/UV irradiation, which could be attributed to their scavenging with ·OH and their VUV screening capacities [22,36]. Li et al. [37] found that NO_3_^−^ may increase the photolysis efficiency of FQs under UV-254 and solar irradiation by producing ROS and nitrogen reactive species (i.e., NO, NO_3_^−^, and N_2_O_4_), but it also suppresses FQs’ photolysis under UV-254 and solar irradiation through competitive photoabsorption with FQs. Ge et al. [17] also proved that the coexisting NO_3_^−^ inhibited the photodegradation of FQs, as it mainly acted as a radiation filter and/or scavenger for ROS. However, they claimed that Cl^−^ had no significant impact on the photodegradation of FQs. Morimura et al. [38] also found that Cl^−^ had no effect on the photodegradation of orbifloxacin. Our results also proved that the impacts of anions on FQs’ photodegradation varied with the types of anions and FQs. The NO_3_^−^ was expected to play a significantly negative role in the degradation of FQs, particularly for NOR. In comparison, no obvious inhibitory effects of Cl^−^ on both NOR and LVF were found.

As depicted in Figure 2, Figure 3, Figure 4 and Figure 5, the FQs’ degradation efficiency varied with the amounts of Pb and Cd salts added. It is likely that the FQ–heavy metal ion complexation altered the molecular orbital components of the excitation and orbital structures, resulting in different light absorption characteristics between FQ–metal complexes and FQ species alone. According to the results in Appendix A, when the molar ratio was 1:1, most of the heavy metal ions could coordinate with FQs, and there were free FQs in the aquatic solution. In comparison, there would be extra metal ions when the molar ratio was 1:3. However, there was no unitary photodegradation performance for all the molar ratios (FQs to heavy metal ions), but a higher molar ratio tends to favor the photodegradation of both NOR and LVF.

The photodegradation mechanisms, products, and pathways of FQs have been studied extensively. As mentioned earlier, FQs can absorb sunlight directly and undergo apparent photolysis. Fe^3+^, Cu^2+^, Ca^2+^, and nitrate can affect the self-sensitized photooxidation rate via changing the properties of hydroxyl radicals (·OH) and ^1^O_2_, and photocatalytic degradation occurs via four pathways: (I) defluorination, (II) hydroxylation in the quinolone ring, (III) dealkylation in the piperazine ring, and (IV) oxidation in the piperazine ring [39,40,41,42,43,44]. For danofloxacin, LEV, difloxacin, and enrofloxacin, N^4^ dealkylation was the main pathway, followed by decarboxylation and defluorination [17]. Wu et al. [45] found that AgI@Ag_3_PO_4_ enabled the photodegradation of norfloxacin and suggested that ^1^O_2_ radicals are the major active species in the visible-light-driven photodegradation of norfloxacin, with sustained attacks on the piperazine ring and the generation of intermediates, while OH radicals are not the major contributors. However, G e et al. [18] suggested that OH can oxidize almost all classes of organic chemicals because of the lower selectivity, oxidized through hydroxylated defluorination and piperazinyl hydroxylation. The photodegradation of antibiotics by different ions, photocatalysts, and adsorbents will produce different intermediates; there were seven products detected during LEV degradation by Mn(VII) [46]. During the photocatalytic degradation of CIP under different pH conditions, more than ten products or intermediates were identified [45,47]. Some studies found that the toxicities first decreased, then increased, and finally decreased, implying the generation of some more toxic intermediates than the parent compound [17]. Some researchers found that solar light effectively degraded FQs, but the photodegradation products retained significant biotoxicity [15,46,48]. However, Zhou et al. [49,50,51] reached the opposite conclusion about photodegradation products. As a result, the photo-modified toxicities for the water ecological environment need more attention. Heavy metal ions can complex with antibiotics and affect the photodegradation, antagonistic, and synergistic effects. Some metal ions serve as a natural photocatalyst in environmental water. Metal complexation can alter the ^1^O_2_ oxidation reaction pathways of antibiotics due to the rearrangement of atomic charges, and also alter the molecular orbital components of the excitation and orbital structures, causing different light absorption characteristics [34]. The oxygen-containing functional groups, namely carbonyl and carboxyl groups, of FQs are able to form complexes with metal ions, which is favorable for the complex compound to enter the triplet state, making the intersystem crossing transitions more efficient [43]. NO_3_^−^ can inhibit the photodegradation of quinolone antibiotics in water by competitively absorbing solar photons or quenching ROS [17,52]. In the future, the degradation behavior and pathways of FQs in real wastewater should be further studied, due to the coexistence of various substances.

## 4. Conclusions

The effects of Pb^2+^, Cd^2+^, Cl^−^, and NO_3_^−^ on the photolytic degradation of NOR and LVF in aqueous solutions were investigated under natural sunlight irradiation. Our results demonstrated that the photodegradation of FQs occurred in aquatic solutions under natural sunlight irradiation. Both NOR and LVF’s photolytic degradation followed the first-order kinetics, and lower initial concentrations of FQs corresponded to higher photolysis efficiency. Both heavy metal ions tended to enhance the FQs’ photodegradation. The effects of Cd^2+^ on the degradation efficiency were more significant than those of Pb^2+^ for both FQs. The presence of aqueous NO_3_^−^ obviously suppressed the NOR degradation, but Cl^−^ had slight effects on these two FQs’ photodegradation.

## Figures and Tables

**Figure 1 ijerph-20-00323-f001:**
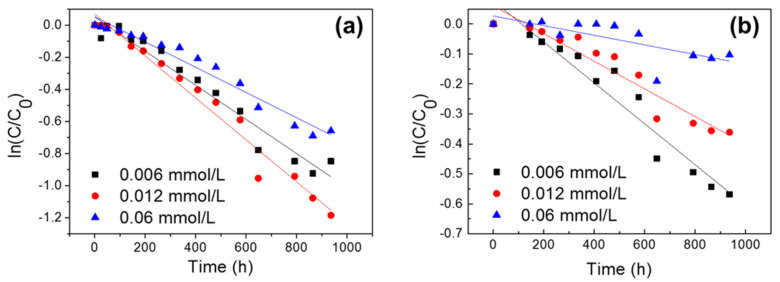
Effects of initial FQ concentration on the attenuation of (**a**) NOR and (**b**) LVF (*C*, the concentration of FQs as a function of reaction time; *C*_0_, the initial concentration).

**Figure 2 ijerph-20-00323-f002:**
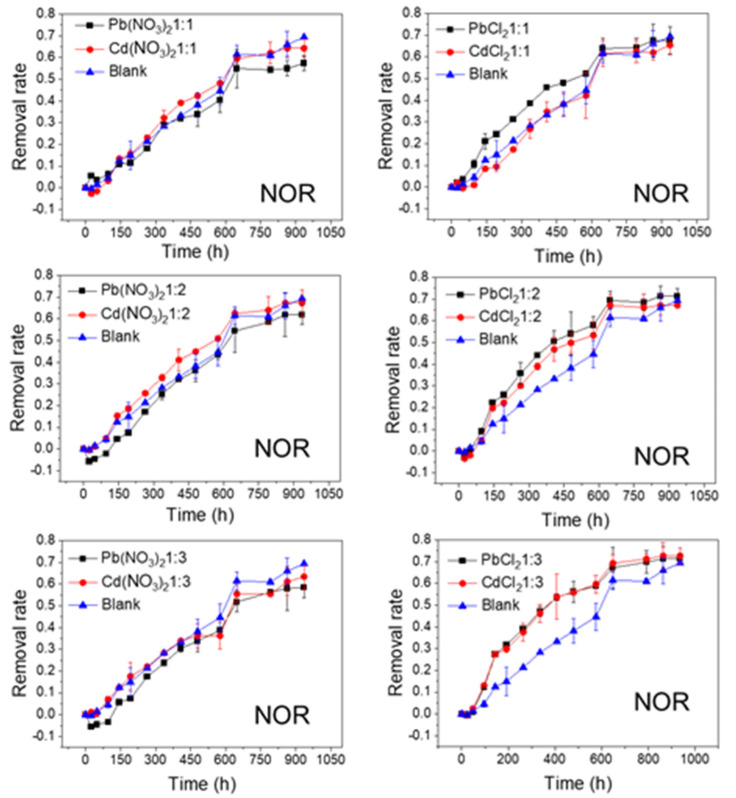
Photolytic degradation of NOR (0.012 mmol/L) with the effects of Pb^2+^ and Cd^2+^ at different molar ratios; Blank means no heavy metal ions added.

**Figure 3 ijerph-20-00323-f003:**
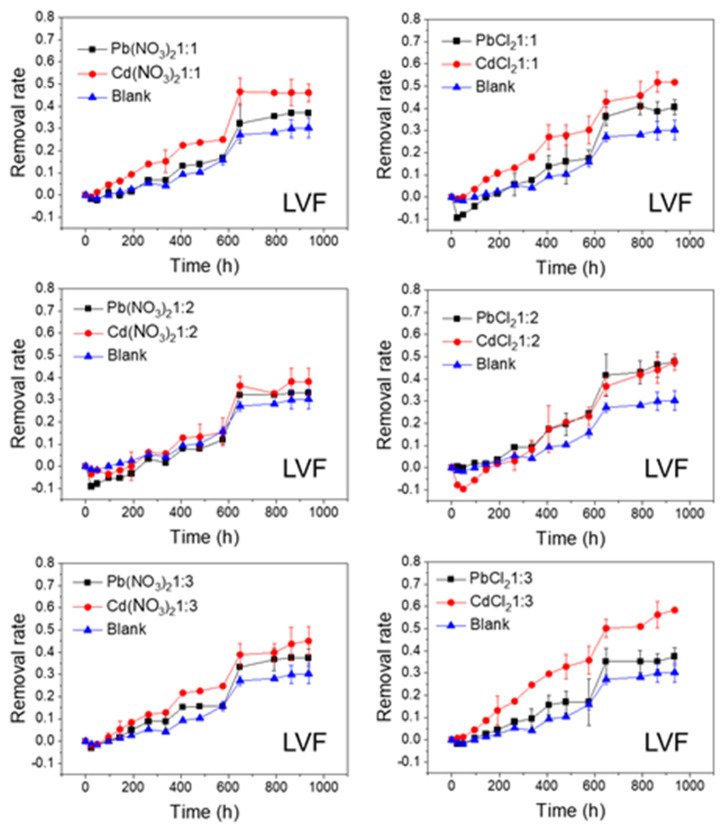
Photolytic degradation of LVF (0.012 mmol/L) with the effects of Pb^2+^ and Cd^2+^ at different molar ratios; Blank means no heavy metal ions added.

**Figure 4 ijerph-20-00323-f004:**
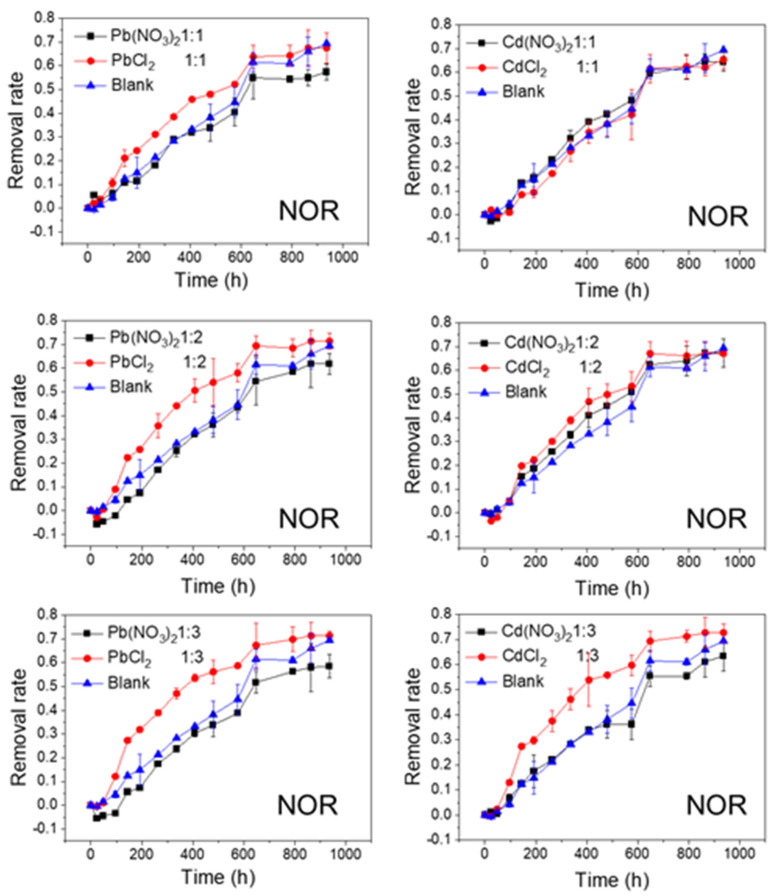
Photolytic degradation of NOR (0.012 mmol/L) with the effects of NO_3_^−^ and Cl^−^ at different molar ratios; Blank means no heavy metal ions added.

**Figure 5 ijerph-20-00323-f005:**
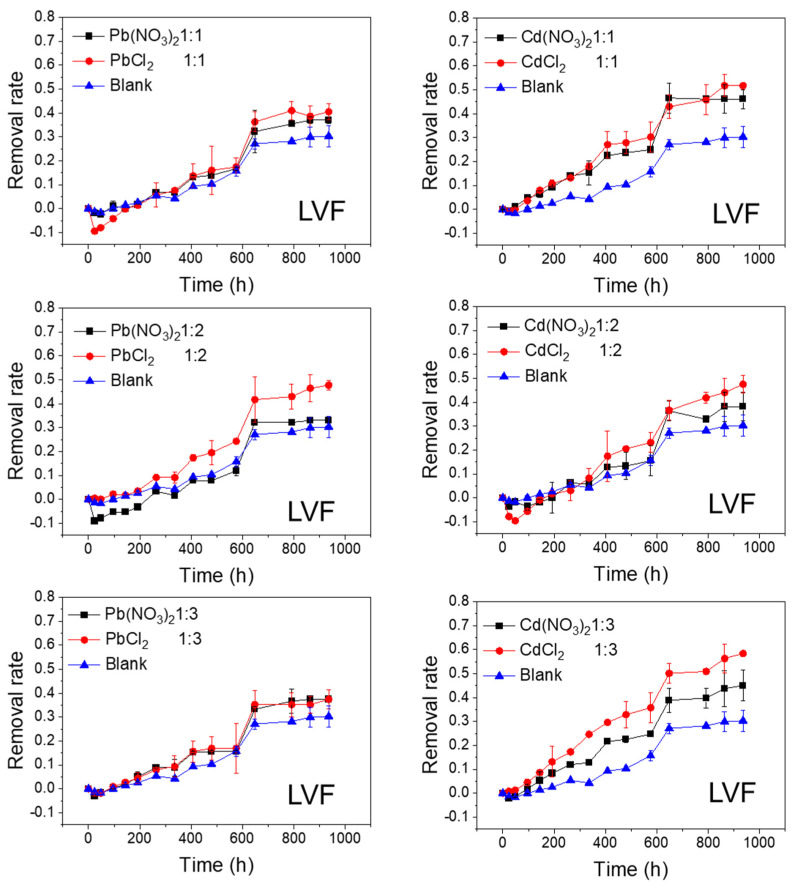
Photolytic degradation of LVF (0.012 mmol/L) with the effects of NO_3_^−^ and Cl^−^ at different molar ratios; Blank means no heavy metal ions added.

## Data Availability

Not applicable.

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
