# Peer review of "Effect of Different Lead and Cadmium Salts on the Photolytic Degradation of Two Typical Fluoroquinolones under Natural Sunlight Irradiation"

_ijerph, 2022, doi:10.3390/ijerph20010323_

Round 1
Reviewer 1 Report
The author tried to study the influence of metal ions on the degradation of FQs under natural photolysis conditions. However, no other valuable data was provided in addition to the detection of FQs concentration, such as the intermediate products or complex structures, which hindered the reasonable analysis of degradation mechanism, lowered its academic value. This made it more like an experimental report rather than a research paper. Therefore, it should be revised thoroughly before it could be accepted for publication in this journal.
Author Response
The photodegradation mechanism, products and pathways of FQs have been studied extensively. As mentioned earlier, FQs can absorb sunlight directly and undergo apparent photolysis. Fe3+, Cu2+, Ca2+ and nitrate, can affect the self-sensitized photooxidation rate via changing the properties of hydroxyl radicals(·OH) and 1O2, and Photocatalytic degradation via four pathways: (I) defluorination, (II) hydroxylation in quinolone ring, (III) dealkylation in piperazine ring, and (IV) oxidation in piperazine ring [39-44]. For danofloxacin, LEV, difloxacin, and enrofloxacin, N4-dealkylation was the main pathway, followed by decarboxylation and defluorination[17]. Wu et al.[45] found AgI@Ag3PO4 enabled photodegradation of norfloxacin and suggested that 1O2 radicals are the major active species in visible-light driven photodegradation of norfloxacin, sustained attacks the piperazine ring and generate the intermediates, while ·OH radicals are not the major contributions. However, G e et al.[18] suggested that ·OH can oxidize almost all classes of organic chemicals because of the lower selectivity , oxidised through hydroxylated defluorination and piperazinyl hydroxylation. The photodegradation of antibiotics by different ions, photocatalysts and adsorbents will produce different intermediates, there were seven products detected during LEV degradation by Mn(VII)[46]. During photocatalytic degradation of CIP under different pH conditions, more than ten products or intermediates were identified [45,47]. There are studies that found the toxicities first decreased, then increased, and finally decreased, implying the generation of some more toxic intermediates than the parent compound[17]. Some researchers found that solar light effectively degraded FQs, but the photodegradation products retained significant biotoxicity[15,46,48]. However, Zhou et al. [49-51] has the opposite conclusion about photodegradation products. As a result, the photo-modified toxicities for the water ecological environment need more attention. Heavy metal ions can complex with antibiotics, and affect the photodegradation, antagonistic and synergistic effects. Some metal ions serve as the natural photocatalyst in the environmental water. The metal complexation can alter the 1O2 oxidation reaction pathways of antibiotics due to the rearrangement of atomic charges, also alter the molecular orbital components of the excitation and orbital structures, causing different light absorption characteristics[52]. The oxygen-containing functional groups, namely carbonyl and carboxyl groups, of FQs are able to form complexes with the metal ions, which is favorable for the complex compound to enter the triplet state, makes the intersystem crossing transitions more efficient[43]. NO3− can inhibit the photodegradation of quinolone antibiotics in water by competitively absorbing solar photons or quenching ROS[17,53]. In the future, degradation behavior and pathways of FQs in the real wastewater should be further studied, due to the coexistence of various substances.
39 Fisher, J.M., Reese, J.G., Pellechia, P.J., Moeller, P.L., Ferry, J.L., 2006. Role of Fe(III), phosphate, dissolved organic matter, and nitrate during the photodegradation of domoic acid in the marine environment. Environ. Sci. Technol. 40, 2200–2205.
40 Grannas, A.M., Pagano, L.P., Pierce, B.C., Bobby, R., Fede, A., 2014. Role of dissolved organic matter in ice photochemistry. Environ. Sci. Technol. 48, 10725–10733.
41 Mack, J., Bolton, J.R., 1999. Photochemistry of nitrite and nitrate in aqueous solution: a review. J. Photochem. Photobiol. A 128, 1–13.
42 Walse, S.S., Morgan, S.L., Kong, L., Ferry, J.L., 2004. Role of dissolved organic matter, nitrate, and bicarbonate in the photolysis of aqueous fipronil. Environ. Sci. Technol. 38, 3908–3915.
43 Heming, Zhang, Xuedan, et al. 2019. Photophysical and photochemical insights of the photodegradation of norfloxacin: The rate-limiting step and the influence of Ca2+ ion.[J]. Chemosphere. 219, 236-242.
44 Si L , Th C , Pd C , et al. 2020. Photocatalytic transformation fate and toxicity of ciprofloxacin related to dissociation species: Experimental and theoretical evidences - ScienceDirect[J]. Water Res. 185, 116286.
45 Wu Z , Yu J , Wang W , et al. 2022. High-performance photodegradation of norfloxacin enabled by AgI@Ag3PO4 nanostructures[J]. J. Alloys Compd. 891:161877-.
46 Zhou Y , Gao Y , Jiang J , et al. 2021. A comparison study of levofloxacin degradation by peroxymonosulfate and permanganate: Kinetics, products and effect of quinone group[J]. J. Hazard. Mater. 403, 123834.
47 Tang L , Wang J , Zeng G , et al. 2016. Enhanced photocatalytic degradation of norfloxacin in aqueous Bi2WO6 dispersions containing nonionic surfactant under visible light irradiation[J]. J. Hazard. Mater. 306, 295-304.
48 Sturini M, Speltini A, Maraschi F, et al. 2016. Removal of fluoroquinolone contaminants from environmental waters on sepiolite and its photo-induced regeneration. Chemosphere. 150, 686–693.
49 Santos LV, Meireles AM, Lange LC. 2015. Degradation of antibiotics norfloxacin by Fenton, UV and UV/H2O2. J Environ Manage. 154, 8–12.
50 Serna-Galvis EA, Ferraro F, Silva-Agredo J, et al. 2017. Degradation of highly consumed fluoroquinolones, penicillins and cephalosporins in distilled water and simulated hospital wastewater by UV254 and UV254/persulfate processes. Water Res. 122, 128–138.
51 Ao X, Liu W, Sun W, et al. 2018. Medium pressure UV-activated peroxymonosulfate for ciprofloxacin degradation: kinetics, mechanism, and genotoxicity. Chem Eng J. 345, 87–97.
52 Wei X , Chen J , Xie Q , et al. 2015. Photochemical behavior of antibiotics impacted by complexation effects of concomitant metals: a case for ciprofloxacin and Cu(II)[J]. Environmental Science: Processes and Impacts. 17(7), 1220.
53 Niu J , Zhang L , Yang L , et al. 2013. Effects of environmental factors on sulfamethoxazole photodegradation under simulated sunlight irradiation:Kinetics and mechanism[J]. J. Environ. Sci. 25(6), 1098-1106.

Reviewer 2 Report
The manuscript of Duan, et al. describes the aqueous photochemical behavior of two typical fluoroquinolones (FQs, such as LVF and NOR) under natural sunlight irradiation. As the authors point out, this is territory that has been tread before by others (Ge et al. 2010, Luo et al. 2019, Ge et al. 2015, Ge et al. 2018, Wei et al. 2015, Wei et al. 2013). Nevertheless, researches on the photolytic degradation of FQs when they interact with heavy metals (Pb2+ and Cd2+) still remains limited and needs further investigation. Importantly, there are relatively fewer photochemical studies of chemicals exposed to the natural sunlight in actual environmental scenarios. The authors have exhibited a good story and presented new findings about the effect of Pb2+/Cd2+ on the photodegradation of the two FQs under natural sunlight irradiation. This work is of high quality and is clearly presented, and will be of interest to the readership. I have just a few comments.
1. In the section 2.2, the reliable determination of NOR and LVF was proved. However, when different lead and cadmium salts coexist with the FQs, and even their complexation occurs, can the accurate quantification of FQs be performed?
2. Many data (from Tables and Figs.) are repetitive. Some data (Table 1) can be put into the Supplementary Materials.
3. In the natural surface waters, coexistence of antibiotics and heavy metals is common. In the introduction, please illustrate their coexistence or concentration levels in natural waters.
4. Based on the findings of the work, heavy metal ions tended to enhance the FQs photodegradation. The corresponding environmental implication or risk should be discussed.
Refs.
Ge, L.K., Chen, J.W., Wei, X.X., Zhang, S.Y., Qiao, X.L., Cai, X.Y. and Xie, Q. (2010) Aquatic photochemistry of fluoroquinolone antibiotics: Kinetics, pathways, and multivariate effects of main water constituents. Environmental Science & Technology 44(7), 2400-2405.
Luo, X., Wei, X., Chen, J., Xie, Q., Yang, X. and Peijnenburg, W.J. (2019) Rate constants of hydroxyl radicals reaction with different dissociation species of fluoroquinolones and sulfonamides: Combined experimental and QSAR studies. Water Research 166, 115083.
Ge, L.K., Na, G.S., Zhang, S.Y., Li, K., Zhang, P., Ren, H.L. and Yao, Z.W. (2015) New insights into the aquatic photochemistry of fluoroquinolone antibiotics: Direct photodegradation, hydroxyl-radical oxidation, and antibacterial activity changes. Science of the Total Environment 527-528C, 12-17.
Ge, L.K., Halsall, C., Chen, C.E., Zhang, P., Dong, Q.Q. and Yao, Z.W. (2018) Exploring the aquatic photodegradation of two ionisable fluoroquinolone antibiotics – gatifloxacin and balofloxacin: Degradation kinetics, photobyproducts and risk to the aquatic environment. Science of the Total Environment 633, 1192-1197.
Wei, X.X., Chen, J.W., Xie, Q., Zhang, S.Y., Li, Y.J., Zhang, Y.F. and Xie, H.B. (2015) Photochemical behavior of antibiotics impacted by complexation effects of concomitant metals: A case for ciprofloxacin and Cu(II). Environmental Science: Processes & Impacts 17(7), 1220-1227.
Wei, X.X., Chen, J.W., Xie, Q., Zhang, S.Y., Ge, L.K. and Qiao, X.L. (2013) Distinct photolytic mechanisms and products for different dissociation species of ciprofloxacin. Environmental Science & Technology 47(9), 4284-4290.
Author Response
Dear Editor,
We sincerely appreciate the editor and the reviewers for their detailed and constructive comments, which stimulate new thoughts and improve our manuscript. Below is a point-to-point list of our responses to the comments and the corresponding changes. The revisions of our manuscript are highlighted in red color for your easy reference. We look forward to hearing from you again soon.
Best regards,
Fenghe Wang (on behalf of co-authors)
Qquestion1. In the section 2.2, the reliable determination of NOR and LVF was proved. However, when different lead and cadmium salts coexist with the FQs, and even their complexation occurs, can the accurate quantification of FQs be performed?
Response: We appreciate the reviewer’s suggestion. In fact, the complexation between heavy metals and antibiotics were generally not strong and easy to dissociate in acidic state (pH <2) after complexing, the samples were treated. Therefore, the determination of antibiotics would not be interfered by heavy metal ions. The quantification was accurate as we checked the controls without the addition of heavy metal every time. Moreover, similar UV-vis spectrophotometry method was used for the determination of anitibiotics in the presence of heavy metals in many previous reports from other academic groups like Prof. Zhou in Insitute of Science, Chinese Academy of Sciences and Prof. Liu of Nanjing University.
Qiu H , Ling C , Yuan R , et al. Bridging effects behind the coadsorption of copper and sulfamethoxazole by a polyamine-modified resin[J]. Chemical Engineering Journal, 2019, 362:422-429.
Qquestion2. Many data (from Tables and Figs.) are repetitive. Some data (Table 1) can be put into the Supplementary Materials.
Response: We appreciate the reviewer’s suggestion. We have revised these parts in the revised manuscript,
Tables 1 and 2 have been put into the supplementary materials.
Supplementary Materials: The following supporting information can be downloaded at: www.mdpi.com/xxx/s1, Table 1. The correlation coefficients (R2), rate constants (k) and half-life time (t1/2) of the two FQs.; Table 2. The pseudo-first-order kinetics parameters of FQs at different molar ratios.; Figure S1: UV spectrum of NOR and LVF. (a) NOR; (b) LVF.; Figure S2: Determination of complexation ratio of NOR/LVF with Pb2+ and Cd2+.
Qquestion3. In the natural surface waters, coexistence of antibiotics and heavy metals is common. In the introduction, please illustrate their coexistence or concentration levels in natural waters.
Response: We appreciate the reviewer’s suggestion.
Relatively high residual FQs concentrations (ng/L to several μg/L) have been widely detected in such as seas [5,6], rivers [7,8], lakes [9,10], groundwater [11,12]and wastewater [13,14], Ciprofloxacin (0 – 3.40 μg /L), norfloxacin (0.037 – 9.35 μg /L ) and ofloxacin (0.047 – 8.64 μg /L) are the most commonly detected quinolone antibiotics in effluents[15], which would potentially pose a significant risk to ecosystems and human health.
With the development of modern agriculture and industry, increasing amounts of heavy metals along with FQs have been continuously discharged into the surface water environments [9,27–29]. As reported in a survey in China, the total amount emissions of heavy metal(loid)s such as lead, cadmium, chromium and arsenic in national wastewater reached 120 tons in 2019. Many studies have shown that rivers, surface water and sewage contain heavy metals, from ng/L~ μg/L, such as Cd(0.025μg/L) in Taihu Lake[30], Pb(1.21 μg/L) in Yangtze River, Pb(0.68μg/L),Cd(0.07μg/L) in urban surface water of SuZhou. Vinod Kumar et al [31] collected 147 publications and proceedings on heavy metals in worldwide surface water bodies, and found that the average concentration of Cd was 180 μg/L. It is necessary to investigate the environmental fate of FQs when they interact with heavy metals, in particular when the complex systems are exposed to the natural sunlight in actual environmental scenarios. Nevertheless, research on this topic still remains limited and needs further investigation.
15 Yang Q, Gao Y, Ke J, et al. 2021. Antibiotics: An overview on the environmental occurrence,toxicity, degradation, and removal methods[J]. Bioengineered. 12: 7376-7416.
30 Zuo, J.X., Fan, w.h., Wang, X.L., et al. 2018. Trophic transfer of Cu, Zn, Cd, and Cr, and biomarker response for food webs in Taihu Lake, China[J]. RSC Advances. 8.
31 Kumar V , Parihar R D , Sharma A , et al. 2019. Global evaluation of heavy metal content in surface water bodies: A meta-analysis using heavy metal pollution indices and multivariate statistical analyses[J]. Chemosphere. 236, 124364.
Qquestion4and5. Based on the findings of the work, heavy metal ions tended to enhance the FQs photodegradation. The corresponding environmental implication or risk should be discussed. The author tried to study the influence of metal ions on the degradation of FQs under natural photolysis conditions. However, no other valuable data was provided in addition to the detection of FQs concentration, such as the intermediate products or complex structures, which hindered the reasonable analysis of degradation mechanism.
Response: We appreciate the reviewer’s suggestion.
The photodegradation mechanism, products and pathways of FQs have been studied extensively. As mentioned earlier, FQs can absorb sunlight directly and undergo apparent photolysis. Fe3+, Cu2+, Ca2+ and nitrate, can affect the self-sensitized photooxidation rate via changing the properties of hydroxyl radicals(·OH) and 1O2, and Photocatalytic degradation via four pathways: (I) defluorination, (II) hydroxylation in quinolone ring, (III) dealkylation in piperazine ring, and (IV) oxidation in piperazine ring [39-44]. For danofloxacin, LEV, difloxacin, and enrofloxacin, N4-dealkylation was the main pathway, followed by decarboxylation and defluorination[17]. Wu et al.[45] found AgI@Ag3PO4 enabled photodegradation of norfloxacin and suggested that 1O2 radicals are the major active species in visible-light driven photodegradation of norfloxacin, sustained attacks the piperazine ring and generate the intermediates, while ·OH radicals are not the major contributions. However, G e et al.[18] suggested that ·OH can oxidize almost all classes of organic chemicals because of the lower selectivity , oxidised through hydroxylated defluorination and piperazinyl hydroxylation. The photodegradation of antibiotics by different ions, photocatalysts and adsorbents will produce different intermediates, there were seven products detected during LEV degradation by Mn(VII)[46]. During photocatalytic degradation of CIP under different pH conditions, more than ten products or intermediates were identified [45,47]. There are studies that found the toxicities first decreased, then increased, and finally decreased, implying the generation of some more toxic intermediates than the parent compound[17]. Some researchers found that solar light effectively degraded FQs, but the photodegradation products retained significant biotoxicity[15,46,48]. However, Zhou et al. [49-51] has the opposite conclusion about photodegradation products. As a result, the photo-modified toxicities for the water ecological environment need more attention. Heavy metal ions can complex with antibiotics, and affect the photodegradation, antagonistic and synergistic effects. Some metal ions serve as the natural photocatalyst in the environmental water. The metal complexation can alter the 1O2 oxidation reaction pathways of antibiotics due to the rearrangement of atomic charges, also alter the molecular orbital components of the excitation and orbital structures, causing different light absorption characteristics[52]. The oxygen-containing functional groups, namely carbonyl and carboxyl groups, of FQs are able to form complexes with the metal ions, which is favorable for the complex compound to enter the triplet state, makes the intersystem crossing transitions more efficient[43]. NO3− can inhibit the photodegradation of quinolone antibiotics in water by competitively absorbing solar photons or quenching ROS[17,53]. In the future, degradation behavior and pathways of FQs in the real wastewater should be further studied, due to the coexistence of various substances.
39 Fisher, J.M., Reese, J.G., Pellechia, P.J., Moeller, P.L., Ferry, J.L., 2006. Role of Fe(III), phosphate, dissolved organic matter, and nitrate during the photodegradation of domoic acid in the marine environment. Environ. Sci. Technol. 40, 2200–2205.
40 Grannas, A.M., Pagano, L.P., Pierce, B.C., Bobby, R., Fede, A., 2014. Role of dissolved organic matter in ice photochemistry. Environ. Sci. Technol. 48, 10725–10733.
41 Mack, J., Bolton, J.R., 1999. Photochemistry of nitrite and nitrate in aqueous solution: a review. J. Photochem. Photobiol. A 128, 1–13.
42 Walse, S.S., Morgan, S.L., Kong, L., Ferry, J.L., 2004. Role of dissolved organic matter, nitrate, and bicarbonate in the photolysis of aqueous fipronil. Environ. Sci. Technol. 38, 3908–3915.
43 Heming, Zhang, Xuedan, et al. 2019. Photophysical and photochemical insights of the photodegradation of norfloxacin: The rate-limiting step and the influence of Ca2+ ion.[J]. Chemosphere. 219, 236-242.
44 Si L , Th C , Pd C , et al. 2020. Photocatalytic transformation fate and toxicity of ciprofloxacin related to dissociation species: Experimental and theoretical evidences - ScienceDirect[J]. Water Res. 185, 116286.
45 Wu Z , Yu J , Wang W , et al. 2022. High-performance photodegradation of norfloxacin enabled by AgI@Ag3PO4 nanostructures[J]. J. Alloys Compd. 891:161877-.
46 Zhou Y , Gao Y , Jiang J , et al. 2021. A comparison study of levofloxacin degradation by peroxymonosulfate and permanganate: Kinetics, products and effect of quinone group[J]. J. Hazard. Mater. 403, 123834.
47 Tang L , Wang J , Zeng G , et al. 2016. Enhanced photocatalytic degradation of norfloxacin in aqueous Bi2WO6 dispersions containing nonionic surfactant under visible light irradiation[J]. J. Hazard. Mater. 306, 295-304.
48 Sturini M, Speltini A, Maraschi F, et al. 2016. Removal of fluoroquinolone contaminants from environmental waters on sepiolite and its photo-induced regeneration. Chemosphere. 150, 686–693.
49 Santos LV, Meireles AM, Lange LC. 2015. Degradation of antibiotics norfloxacin by Fenton, UV and UV/H2O2. J Environ Manage. 154, 8–12.
50 Serna-Galvis EA, Ferraro F, Silva-Agredo J, et al. 2017. Degradation of highly consumed fluoroquinolones, penicillins and cephalosporins in distilled water and simulated hospital wastewater by UV254 and UV254/persulfate processes. Water Res. 122, 128–138.
51 Ao X, Liu W, Sun W, et al. 2018. Medium pressure UV-activated peroxymonosulfate for ciprofloxacin degradation: kinetics, mechanism, and genotoxicity. Chem Eng J. 345, 87–97.
52 Wei X , Chen J , Xie Q , et al. 2015. Photochemical behavior of antibiotics impacted by complexation effects of concomitant metals: a case for ciprofloxacin and Cu(II)[J]. Environmental Science: Processes and Impacts. 17(7), 1220.
53 Niu J , Zhang L , Yang L , et al. 2013. Effects of environmental factors on sulfamethoxazole photodegradation under simulated sunlight irradiation:Kinetics and mechanism[J]. J. Environ. Sci. 25(6), 1098-1106.
.
